# A Banking Platform to Leverage Data Driven Marketing with Machine Learning

**DOI:** 10.3390/e24030347

**Published:** 2022-02-28

**Authors:** Marc Torrens, Amir Tabakovic

**Affiliations:** 1Department of Operations, Innovation and Data Sciences, ESADE Business School, University Ramon Llull, 08172 Sant Cugat del Vallès, Spain; 2Institute Applied Data Science and Finance, Business School, University of Applied Sciences, 3005 Bern, Switzerland; amir.tabakovic@bfh.ch

**Keywords:** banking, payment data, machine learning, marketing

## Abstract

Payment data is one of the most valuable assets that retail banks can leverage as the major competitive advantage with respect to new entrants such as Fintech companies or giant internet companies. In marketing, the value behind data relates to the power of encoding customer preferences: the better you know your customer, the better your marketing strategy. In this paper, we present a B2B2C lead generation application based on payment transaction data within the online banking system. In this approach, the bank is an intermediary between its private customers and merchants. The bank uses its competence in Machine Learning driven marketing to build a lead generation application that helps merchants run data driven campaigns through the banking channels to reach retail customers. The bank’s retail customers trade the utility hidden in its payment transaction data for special offers and discounts offered by merchants. During the entire process banks protects the privacy of the retail customer.

## 1. Introduction

In the last ten years, the banking industry has faced rising pressure by new challengers, particularly FinTech companies, telecommunication operators and technology giants in business areas like payments, money management and lending.

The increasing immersiveness of digital technologies in our daily life, commoditization of computing and storage capacities, advances in crypto technologies and the area of artificial intelligence and new regulations, like PSD2 [1] in Europe that is forcing the banks to open their data and services to third party providers, have lowered entry barriers for new players to question the position of banks on their home turf.

Banks are not only vulnerable to the innovators that focus on efficiency improvements of existing financial services. Even the unfavorable position of a “dumb pipe” that only processes payments while leaving the high margin business to other market players is under pressure by the emerging cryptocurrencies [2].

The financial service industry has reacted to the new threats by different measures such as: increasing its investments in new technologies, partnering with FinTech companies or creating industry alliances to name a few. Still most of the efforts are reactive by nature and focus in trying to defend the existing business models. Banks are protecting the customer interface, the existing payment system, their position in the value chain. It seems like banks are suffering under the endowment effect and loss aversion when it comes to business model innovation. Innovation is welcome as long as it happens within the boundaries of the existing traditional banking business model.

### 1.1. Payment Transactions, the Precious Data Asset

Transactional data is one of the most valuable assets that banks can leverage as the major competitive advantage with respect to new entrants such as Fintech companies or giant internet companies. In advertising, the value behind data relates to the power of encoding customer preferences: the best you know your customer the better is your marketing strategy.

Most of the online advertising, is based on behavioral data and aspirational data. Customer preferences are based on customer online behavior. However, it is being shown that online behavior only encodes a partial view on the customer. On the other hand, transactional data encodes in a very concise manner our daily life: “tell me how you spend, and I will tell you who you are”. To illustrate this concept, just imagine a customer loving fine dining, regularly watching Master Chef’s Youtube channels, visiting all the websites about haute cuisine, and following many Instagram fine dining accounts. His online behavior shows that he loves fine dining. However, his spending behavior tells a different story. He purchases food in budget supermarkets and visits only fast-food restaurants. Only the transactional data encodes his true spending behavior, so it is a very valuable information that other online activity cannot show. In general, we could conclude that online behavioral data encodes the aspirational view on the customer, while transactional data encodes the reality of the customer.

The other unique characteristic of customer data in banks when compared to other customer data is the historical depth of transactional data. Banks often store the entire financial transaction history of a customer and that could also give new perspective on how customer’s preferences change through time: “tell me who you were, and I will tell you who you are”. Omer Artun [3] analyzes different approaches to predict what consumers will buy and concludes that past purchase data is more meaningful than aspirational or intentional data in most of the scenarios.

Banks have an enormous opportunity to create new business models based on the data they have been collecting for so many decades [4]. When comparing banking data with advertising data from the web, banking data has the particularity of encoding the reality of our daily life. Customer’s current account is his diary.

This understanding of the customer will allow banks to deliver the personalization the customer and merchants are looking for.

### 1.2. Personalization on the Web

Advertisement is one of the key business drivers for most of the companies operating on the internet, but an unexplored venue in online banking. In [5], Puschmann is describing the customer-oriented digitalization for banks around wallets including not only payment, but also the option to collect, store and spend loyalty points and other personal data. This project aims at exploring advertisement with commercial owners within online banking applications.

The degree of personalization for advertisement on the web strongly depends on the type and sophistication level of the platform. For example, in search engines such as Google, personalized advertisement is based on more than 50 factors such as location, day of the week, time of the day, accessing device, browsing history, and so on. Advertisement on newspapers is usually personalized based on the displayed content. Additionally, in web applications, such as Spotify, Mint or Youtube, advertisement is usually personalized based on user profiles and their specific activity within the application. On the other hand, companies such as Magnite (See https://www.magnite.com/ (accessed on 6 January 2022)) are revolutionizing online advertisement [6] with high frequency trading online platforms that operate across several websites in real time. Nowadays, the advertisement model is being fully exploited on the internet. However, within online banking applications this is a venue to be yet analyzed and implemented for most of the retail banks. There are many reasons for having online banking applications free of advertisement, mostly related to the fact that advertisement could damage the perception of the bank and distract customers from its main goal which is to operate efficiently with the bank. Despite those arguments, we believe offering tailored commercial recommendations should be interesting for retail customers and provide value to the bank, especially if offers are displayed as recommendations and not as traditional commercial advertisement [7]. Usability plays an important role to make that distinction between commercial advertisement and personalized recommendations.

### 1.3. Lead Generation Application for Online Banking Systems

In this paper, we describe an application within the online banking system to leverage the transactional data value in matching the right merchants to customers. In the following section, the business architecture is described. The rest of the paper is devoted to the data modeling and Machine Learning techniques in order to target merchants’ offers according to consumers’ spending habits.

## 2. The Business Architecture

The key factors for recommending commercial offers from corporate customers (merchants) to retail customers (consumers) within an online banking environment are:Commercial recommendations can be done based on transactional data that encode spending behavior in addition to the demographic features of the retail customer. This allows us to implement sophisticated personalization algorithms. In other words, the data from the end user that are available in a bank environment is extremely rich compared to the data that is available in other internet domains.The bank is in full control of the whole transactional loop as depicted in Figure 1: corporate customers, retail customers, and transactional data resides all in the same system. This close but comprehensive environment allows the bank to build the whole system being in control of the rich data for retail customers and corporate customers, implementing the whole internet application, and executing a robust business model.

The business architecture is depicted in Figure 1 with the following steps:In step 1, the corporate customer defines and uploads the commercial offer together with a commercial strategy and a defined audience. Characteristics of these commercial offers are described later on.In the second step, our algorithms described in Section 4.1 order retail customers for each commercial offer available in the system. This ranking is done through a set of metrics and Machine Learning models. The objective is to find retail customers that are interested in those particular offers.Retail customers receive the offers in their online banking system (i.e., mobile banking app) and decide if they want to accept them. Accepting an offer is a necessary step to benefit from them. This step is necessary to avoid customers benefiting from offers by chance.After the customer has accepted the offer, the transaction can happen. One of the advantages of this system is that the purchase is done as usual with a credit or debit card. No special coupons, or credentials, or specific software is needed. The purchase is happening as any other electronic transaction.Step 5 is about charging the offers that have been redeemed to corporate customers (merchants). This is happening in the background of the banking system, and in one step at the end of the month, collecting all the redeemed offers.Similarly to step 5, in this step the bank collects all the redeemed offers from the system and sends the money of the redeemed offers to the retail customer. This step is also happening at the end of the month, in the background.

On the business model side, different revenue sources can be implemented, namely:Annual subscription fees paid by the participant corporate customers. Merchants would pay a fee based on the service they are getting within the online banking system.Impression fees paid by the corporate customer whose offer is displayed. Merchants would pay based on a success fee proportional to the number of displayed offers to retail customers.Redemption fees paid by the corporate customer whose offer is redeemed. In this case, merchants would pay a fee proportional to the number of redeemed offers.

The goal of the personalization algorithms is to increase the redemption rates so that the loyalty program is proven to be efficient and useful for both its corporate customers and its retail customers. System efficiency in offer personalization would have a direct impact on the business model.

## 3. Preliminaries

In this section, we formally define some key concepts for the described system, namely: audiences, commercial opportunities and campaigns.

### 3.1. Audiences

An Audience is a set of retail customers that satisfy a set of filtering criteria. In traditional marketing, audiences are usually defined through demographic filtering criteria. However, in our scenario, we are interested in defining target customers based on their financial behavior encoded in their transactional data. The overall goal is to target customers based on their commercial interests. We are assuming that transactional data from customers decode precisely commercial interests that we want to consider in the targeting process.

Formally, an Audience consists of a set of filtering criteria classified in commercial interests and demographic aspects. Formally, consider *U* the set of users *u*, which are retail customers of the bank ($\left|U\right|=1M$), and a set of filtering criteria as Boolean functions F={f:u→B}. Then, formally, an audience is defined as: $O=\{u:f_{k}\left(u\right)\}\subseteq U$.

Those filtering criteria are defined as *Commercial Interests*. We have identified four dimensions defining Commercial Interests that are relevant to create audiences, namely Loyalty, Frequency, Purchase Segmentation and Location. These criteria identify the commercial interest of the retail customer for a given commercial category *c* and merchant *m* in a given period of time *t*. Commercial categories are based on the Merchant Category Codes (See https://en.wikipedia.org/wiki/Merchant_category_code (accessed on 6 January 2022)) (MCC), which define spending categories in retail financial services.

A merchant *m* is always associated with an MCC. We are mapping MCC to a higher category *c*. Thus, we are associating each merchant working for the bank to a spending category *c*. A merchant is defined as the tuple {c,l} where *c* is a category from its MCC and *l* is its location.

On the other hand, an electronic payment transaction is defined as the tuple:Purchase date *t*,Amount *a*,Category *c*,Merchant *m*, andRetail customer.

In the following, we describe those commercial interests that will be used to define audiences for a given Campaign.

**Loyalty** is a very interesting commercial criterion in order to target commercial offers. In some cases, the merchant may want to acquire new customers, in some other cases the merchant may want to reinforce loyalty of existing customers. In general, we define 4 types of loyalty for a given merchant *m* of a spending category *c* in a given period of time *t*:**Loyal customers**: are retail customers who purchase items in the category *c* with a certain high frequency to the merchant *m*.**Shared customers** are retail customers who purchase items in the category *c* in the merchant *m* and other merchants offering items in the category *c*.**Competitors’ customers** are retail customers who buy items in the category *c* but in merchants different than *m*.**Non-customers** are retail customers who do not buy items from the category *c*.**Frequency** is a criterion that allows the system to classify retail customers into frequency categories such as high, medium and low related to a spending category *c*. In this way, a merchant can target an offer to people that buy items in its category depending on their buying frequency. Note that this criterion is relative to each category and different levels can be defined.**Purchase Segmentation** is a criterion that describes the average amount *a* for a given category *c* spent by the retail customer. In other words, this criterion defines how much the retail customer is spending for a transaction in a given category *c*. Note that this allows the merchant to target its offers to people depending on the average amount of their transactions in that category. In other words, the merchant can target people that buy expensive or inexpensive items within a certain category. Note that this criterion is relative to each category and different levels can be defined.**Location** is a very important filtering criterion to define an opportunity based on the locations given by the transactions of the retail customers. Another filtering criterion could be the location where people live, but we are more interested in the behavioral. This is relevant since a user could live in one area, but do most of the shopping in another (for example, close to their working place). Similarly, this criterion should be defined with different levels (near, far away) related to a given merchant.

### 3.2. Commercial Opportunities

Formally, we define the filtering criteria based on commercial interests that are used to define Commercial Opportunities:

**Loyalty**: In some cases, the merchant may want to acquire new customers, in some other cases the merchant may want to reinforce loyalty of existing customers. In general, we define four types of loyalty for a given merchant *m* of a spending category *c* in a given period of time *t*. Let us first define trx(u,m,t) as the set of transactions from user *u* in merchant *m* in the period of time *t*, and trx(u,c,t) as the set of transactions from user *u* in the spending category *c* in the period of time *t*. For a given merchant *m*, we define its spending category as cat(m).

**Loyal** customers are those customers that mostly buy in *m* for the category *c* in the given time *t*.Formally, $f_{loyalty}$ is a Boolean function defined as:
$$f_{loyalty}\left(u,m,t\right)=\left(\frac{\left|trx(u,m,t)\right|}{\left|trx(u,cat(m),t)\right|}\geq k\right)\wedge \left(\left|trx(u,cat(m),t)\right|>0\right),$$
where *k* is a loyalty threshold.**Shared** customers are those customers that buy sometimes in *m* and sometimes in other merchants for the category *c* in the given time *t*.Formally, $f_{shared}$ is a Boolean function defined as:
$$f_{shared}\left(u,m,t\right)=\left(l\leq \frac{\left|trx(u,m,t)\right|}{\left|trx(u,cat(m),t)\right|}<k\right)\wedge \left(\left|trx(u,cat(m),t)\right|>0\right),$$
where *l* is a competitor threshold.**Competitor** customers are those customers that mostly buy in other merchants than *m* for a the category *c* in the given time *t*.Formally, $f_{competitors}$ is a Boolean function defined as:
$$f_{competitors}\left(u,m,t\right)=\left(0\leq \frac{\left|trx(u,m,t)\right|}{\left|trx(u,cat(m),t)\right|}<l\right)\wedge \left(\left|trx(u,cat(m),t)\right|>0\right)$$**Non-customers** are those customers that made no purchase in category *c* during the period *t*.Formally, $f_{non-customers}$ is a Boolean function defined as:
$$f_{non-customers}\left(u\right)=\left(\left|trx(u,cat(m),t)\right|=0\right)$$

For example, let’s consider k=0.6 and l=0.1, then:Loyal customers are the ones that make at least 60% of their transactions for cat(m) in *m*.Shared customers are the ones that make between 10% and 60% of their transactions for cat(m) in *m*.Competitors customers are the ones that make less than 10% of their transactions for cat(m) in *m*.

Note that the loyalty criterion could also be defined in terms of amount of money spent in the merchant rather than number of transactions. However, we believe the number of transactions in a merchant gives a more accurate measure of the loyalty concept. We define the amount associated with a transaction *t* as amt(t). Then, the loyalty concept related to amounts instead of number of transactions could be implemented simply by replacing:$$\left|trx(u,m,t)\right|by\sum_{i=0..n}amt(t_{i}),t_{i}\in trx\left(u,m,t\right)$$
and
$$\left|trx(u,cat(m),t)\right|by\sum_{i=0..n}amt(t_{i}),t_{i}\in trx\left(u,cat\left(m\right),t\right)$$

**Frequency** is a commercial interest to classify people based on their buying frequency within a given category or merchant. For now, we are only focusing at the category level, but it could also be applied at the merchant level. We are defining this criterion in three different levels (high, medium, low), but other granularity could be exploited.

In a period of time *t*, we will aim at clustering people in three segments or clusters with respect to the number of transactions of people in a category *c*. Formally, for a given person *u*, category *c*, and a period of time *t*, we define $x\left(u,c,t\right)=\left|trx(u,c,t)\right|$. For simplicity we will note x(u,c,t) as x(u) when the category and the period of time are considered constant, e.g., when comparing customers in the same category during the same time period. Thus, we aim at partitioning people into 3 sets $S=\{S_{1},S_{2},S_{3}\}$, and for that purpose we will use the *k-means* method of clustering with k=3. The k-means method then minimizes the within-cluster sum of squares:$$\underset{S}{\mathrm{error}}\sum_{i=1}^{3}\sum_{u_{j}\in S_{i}}{\left|\left|x\left(u_{j}\right)-\mu_{i}\right|\right|}^{2},\mathrm{where}\mu_{i}=\bar{x(u)},u\in S_{i}.$$

We are using the Mahout’s implementation of k-means algorithm to compute frequency clusters (https://cwiki.apache.org/MAHOUT/k-means-clustering.html (accessed on 6 January 2022)).

Purchase Segmentation is another commercial interest to classify people based on their mean purchase amount among transactions for a given category or merchant. The rationale is exactly the same as the above criterion, but replacing the number of transactions by their mean amount. Thus, defining
$$x\left(u,c,t\right)\mathrm{as}\;\bar{amt(trx(u,c,t))}$$
we can use the same method as with the frequency criterion.

Demographic filtering criteria are classic in marketing research and do not need special computation, but just filter the data according to that criteria. We are referring to criteria such as location, gender, age, family status, profession, etc. Note that location as a demographic attribute is different from the location commercial interest. Demographic location refers to where the person lives, and location from a commercial interest point of view refers to the location where the person does shopping. We are not using demographic attributes to define audiences since we are interested in data that defines the actual commercial interest of retail customers that is encoded in their transactional data.

A Commercial Opportunity is an Audience with specific commercial interests. We have identified the following interesting commercial opportunities for the system:**Increase Loyalty**: opportunity for a specific merchant defines an audience of retail customers that are not highly loyal to that merchant. It is proposed for those merchants that want to increase the loyalty to them.**Win New Customers**: opportunity for a specific merchant defines an audience of retail customers that are not regular customers of that merchant, i.e., customers that buy in the merchant category, but not in that merchant. It is proposed for those merchants that want to win new customers.**Increase Frequency**: opportunity for a specific merchant defines an Audience of retail customers that do not purchase very frequently for that category. Please note that this opportunity may refer to customers of that merchant or customers of other merchants for that category.**Increase Spending**: opportunity for a specific merchant defines an audience of retail customers that buy inexpensive products in the category. Thus, the goal of this opportunity is to propose offers of expensive products to customers that usually buy inexpensive products within that category.**Keep Customers**: opportunity for a specific merchant defines an audience of retail customers that are loyal to that merchant. The goal of this opportunity is to target those loyal customers. Those opportunities may be selected by the merchant when defining a Campaign. They may be used as is or as a starting point to define an audience.

### 3.3. Campaigns

A **Campaign** consists of:A **Merchant** publishing the Campaign.An **Audience** that defines the total target group of retail customers.An **Offer** consisting of:-A discount type which is either percentage or absolute number.-A discount that can be either a percentage or an absolute amount of money to be discounted. It is important to note that discounts are not applicable to a specific item sold by the merchant, but to any transaction with the merchant.-A maximum/minimum amount transaction to apply the discount.-A title of the offer.-A descriptions of the offer (can be specified for multiple languages).-A banner of the offer.-A link to the offer or merchant.A start date and an end date defining the period of time for which the offer is valid.A maximum number of impressions the merchant customer is willing to afford.A maximum cost of redemption the merchant customer is willing to afford.A maximum total cost of the Campaign.

## 4. The Algorithm: Targeting Campaigns to Retail Customers

The main purpose of the matching algorithm is to compute a set of *good* campaigns for a given user in real time maximizing the success of all active campaigns. There are multiple factors to consider when evaluating the best campaigns for a given user. Some of these factors relate directly to the level of accomplishment of the active campaigns and their time left. Other factors relate directly to the level of interest of the user for those campaigns. The algorithm schema described below allow us to compute the set of *best* campaigns for a given user in real time considering a number of different factors. The weights of those different factors can be changed in order to adapt the algorithm to specific business goals. The algorithm schema is also flexible enough to incorporate more factors than the ones considered in this description.

### 4.1. Algorithm Schema

The algorithm schema consists of four main steps. The first step computes a data structure to consider the factors related to the active campaigns themselves, independently of the input user. Thus, this first step can be computed regularly (for example once a day) and be applied for all different users logging into the system. The second step is to compute a data structure to consider the factors related to the level of interest of the user to all active campaigns. The level of interest of the user for a Campaign depends on a number of factors as described below. The third step is to combine the data structures in the previous steps into a final data structure that models the overall relevance of a Campaign for the given user. Finally, the last step simply selects the good campaigns for the given user. This algorithm schema allows us to fine tune different parameters or weights to come up with the right factor weights that will result in an optimum algorithm. These weights may be determined by A/B testing or with more analytical processes.

### 4.2. Step 1: Campaign Salience

This step computes the Campaign Salience (OS) for every active Campaign $o_{i}\in O_{A}$. The Campaign Salience $OS(o_{i})$ is computed considering the ratio of accomplishment and the time left for that Campaign. An active Campaign $o_{i}$ has the following parameters:$startDate(o_{i})$ and $endDate(o_{i})$ defining the start and end dates of the active period of the campaign $o_{i}$, and$redemptionsTarget(o_{i})$ and $redemptionsActual(o_{i})$ defining and target number of redemptions for $o_{i}$ and the total number of executed redemptions for $o_{i}$ up to now.

The time ratio gone for an campaign is defined as:$$TR\left(o_{i}\right)=\frac{today-startDate(o_{i})}{endDate-startDate(o_{i})}$$

Analogously, the accomplishment ratio for an active campaign is defined as:$$AR\left(o_{i}\right)=\frac{redemptionsActual(o_{i})}{redemptionsTarget(o_{i})}$$

The Campaign Salience is then defined as $OS\left(o_{i}\right)=TR\left(o_{i}\right)-AR\left(o_{i}\right)$. If $TR\left(o_{i}\right)-AR\left(o_{i}\right)$ is positive, the Campaign is behind its target, otherwise the Campaign is ahead of its target. The idea is that those campaigns with positive $OS(o_{i})$ should be pushed higher to get closer to the final target. The campaigns that are ahead of their target could be pushed lower with respect to the rest.

#### 4.2.1. The Second Step: User-Centered Campaign Salience

This step computes the User-centered Campaign Salience $UOS(o_{i},u_{j})$, namely the salience of a Campaign $o_{i}$ with respect to a specific user $u_{j}$. The UOS function estimates the degree to which a Campaign $o_{i}$ might be of interest to the a specific user $u_{j}$. We consider the following criteria to be taken into consideration for the User-centered Campaign Salience, where [0, 1] is the interval of real numbers between 0 and 1.

**Likelihood** of buying in the category of the Campaign. This function estimates the predicted probability of the user $u_{j}$ to buy in the category $cat(o_{i})$ of the Campaign $o_{i}$ proposed by the merchant $m(o_{i})$ in the following *k* months in which the Campaign will be active (noted as $k=\tau (o_{i})$ henceforth). In this way, we are taking into consideration the odds of the user to be interested in buying in the category of the merchant of the given Campaign.
$$LK\left(u_{j},m\left(o_{i}\right),\tau \left(o_{i}\right)\right)\to \left[0,1\right]$$This function is implemented through the algorithm described in Section 5.**Proximity** of the user $u_{j}$ to the merchant *m* of the Campaign. This function estimates the distance between two locations: (1) the location where the user lives and (2) the merchant’s location. Currently, distance between locations is based on zip codes. We are also applying an exponential decay function to emphasize that longer distances should be penalized much more rapidly. We define $d(u_{j},m\left(o_{i}\right))$ as the distance in Km from the zip code of the user $u_{j}$ to the zip code of the merchant $m(o_{i})$ offering $o_{i}$. Please note that this Proximity function does not consider Internet companies as merchants. For Internet companies the Proximity function should be set up to a certain constant value. In this system, we were only considering small and medium retailers excluding Internet companies. Additionally, proximity could take into consideration the density of the population of a city. In this way, the distance of the customer to the merchant could be made proportional to the density of the city of the customer. Distances in low density cities should be considered less relevant than the same distances in high density cities.
$$PX\left(u_{j},m\left(o_{i}\right)\right)=e^{-d(u_{j},m\left(o_{i}\right))/\kappa }\to \left[0,1\right]$$
where κ is the decay parameter. At κ=100, a distance of 1 km produces a PX of 0.99, a distance of 5 km produces a PX of 0.95, a distance of 10 km produces a PX of 0.90, a distance of 50 km produces a PX of 0.60, a distance of 100 km produces a PX of 0.36, a distance of 300 km produces a PX of 0.049, a distance of 400 km produces a PX of 0.018, etc.**Activity** of the user $u_{j}$ with the merchant $m(o_{i})$ that is making the Campaign $o_{i}$. The ACT function estimates the relevance of the activity of user $u_{j}$ to the merchant $m(o_{i})$ as a whole. In other words, ACT expresses how much business the given user is making with the merchant compared with the rest of the merchant’s customers. ACT is defined as a linear combination of two ratios: one based on the number of transactions and the other based on the spending amount. The two ratios are: (a) the ratio of the number of transactions (trx) by the user compared to the number of transactions by the user that has more transactions in that merchant ($v_{T}$), and (b) the ratio of the total amount of money spent (amt) by the user compared to the total money spent by the user that has spent more in that merchant ($v_{M}$).
$$ACT\left(u_{j},m\left(o_{i}\right)\right)=\left(\alpha_{1}\;\frac{\left|trx(u_{j},m\left(o_{i}\right))\right|}{\left|trx(v_{T},m\left(o_{i}\right))\right|}+\left(1-\alpha_{1}\right)\;\frac{amt(u_{j},m\left(o_{i}\right))}{amt(v_{M},m\left(o_{i}\right))}\right)\to \left[0,1\right]$$
where the set of transactions of the merchant is $V(m\left(o_{i}\right))$, the maximum transaction number is $v_{T}=\underset{v\in V(m\left(o_{i}\right))}{\mathrm{error}}\left(\left|trx(v,m\left(o_{i}\right))\right|\right)$ and the maximum spender is $v_{M}=\underset{v\in V(m\left(o_{i}\right))}{\mathrm{error}}\left(amt\left(v,m\left(o_{i}\right)\right)\right)$**Loyalty** of the user to the merchant of the Campaign. The LY function estimates the loyalty level of the user $u_{j}$ to the merchant $m=m(o_{i})$ for a given category $c=cat(o_{i})$. Basically, LY defines the ratio of activity of the user with the merchant compared with the users activity in the category *c*. LY combines two aspects of loyalty, one related to the number of transactions, and the other one related to the amount money spent:
$$LY\left(u_{j},m\left(o_{i}\right)\right)=\left(\alpha_{2}\;\frac{\left|trx(u_{j},m\left(o_{i}\right))\right|}{\left|trx(u_{j},cat\left(o_{i}\right))\right|}+\left(1-\alpha_{2}\right)\;\frac{amt(u_{j},m\left(o_{i}\right))}{amt(u_{j},cat\left(o_{i}\right))}\right)\to \left[0,1\right]$$**Merchant Fitness** with respect to a user $u_{j}$ considering the median of the merchant’s selling prices. This function estimates how close (1) the median of the user $u_{j}$ transaction amounts in the given category $cat(o_{i})$ is to (2) the median of the merchant *m* transaction amounts:
$$MF\left(u_{j},m\left(o_{i}\right)\right)=\left(1-\frac{\left|M\left(u_{j},cat\left(o_{i}\right)\right)-M\left(m\left(o_{i}\right)\right)\right|}{max(M\left(u_{j},cat\left(o_{i}\right)\right),M\left(m\left(o_{i}\right)\right)}\right)\to \left[0,1\right]$$
where M is the median.

Thus, the User-centered Campaign Salience (UOS) of a Campaign $o_{i}$ for a given user $u_{j}$ is a linear combination of the above factors:$$\begin{array}{c} UOS\left(u_{j},o_{i}\right)=\omega_{1}\;LK\left(u_{j},o_{i},\tau \left(o_{i}\right)\right)+\omega_{2}\;PX(u_{j},m\left(o_{i}\right)+\omega_{3}\;ACT\left(u_{j},m\left(o_{i}\right)\right)+\\ +\omega_{4}\;LY\left(u_{j},m\left(o_{i}\right)\right))+\omega_{5}MF\left(u_{j},m\left(o_{i}\right)\right) \end{array}$$
where $\omega_{1}+\omega_{2}+\omega_{3}+\omega_{4}+\omega_{5}=1$.

#### 4.2.2. The Third Step: Overall Salience

This third step determines the overall salience of a Campaign $o_{i}$ to a user $u_{i}$ by combining the User-centered Campaign Salience (UOS) with the the Campaign Salience (OS) as follows:$$Salience\left(u_{j},o_{i}\right)=\alpha_{3}\;UOS\left(u_{j},o_{i}\right)+\left(1-\alpha_{3}\right)\;OS\left(o_{i}\right)$$

#### 4.2.3. The Fourth Step: Selecting Campaigns

The final step requires selecting *k* campaigns for a given user *u*. The salience of a Campaign $o_{i}$ for a user $u_{j}$ determines the probability of that Campaign to be selected.

Instead of just taking the *k* campaigns with higher Salience for a user, we use inverse transform sampling (ITS, also known as inversion sampling or Smirnov transform.). ITS performs weighted randomized selection; that is to say, ITS selects one of the campaigns probabilistically, assuring that the higher the salience the higher the probability of being selected. This procedure insures *variety* for the user since, no matter how often she logs in the system, the campaigns selected will not be deterministically repeated (which would be the case when nothing changes the factors and the selection just takes the *k* campaigns with highest salience). Selecting *k* campaigns can be achieved simply by repeating *k* times the ITS procedure.

## 5. Predicting Purchase Behavior

In Section 4.2.1, we defined the User-centered Campaign Salience (UOS) considering a function (LK) that estimates the predicted probability of the user $u_{j}$ to buy in the Campaign $o_{i}$ proposed by the merchant $m(o_{i})$ in the following *k* months in which the Campaign will be active (noted as $k=\tau (o_{i})$ henceforth). In this section, we analyze different algorithms to compute this function.

We aim at predicting the odds of a retail customer to purchase in a given category in the following months. We distinguish three different types of data that can be relevant for such prediction:Past transactional data of the user $u_{j}$. Past transactional data for a user should indicate the *generic interest* of a user in a specific category $cat(o_{i})$ in the future, therefore also a degree of interest in a Campaign within a category.Past behavior of $u_{j}$ regarding other campaigns. Past behavior of a user in other campaigns should also indicate something about the degree of interest of a user in accepting a Campaign within a specific category or from a specific merchant. For now, we are not considering users’ feedback, since this data will only be available once the platform is running.Demographic data from the user $u_{j}$. Some demographic attributes could have an impact on the likelihood of a user to accept a Campaign within a specific category or from a specific merchant.

*Supervised Machine Learning* techniques can be applied to learn a model that predicts the likelihood of a user to purchase in a category. Supervised learning is the machine learning task of inferring a function from labeled training data. The training data consist of a set of training examples. In supervised learning, each example is a pair consisting of an input object (typically a vector) and a desired output value (also called the supervisory signal). A supervised learning algorithm analyses the training data and produces an inferred function, which is called a classifier (if the output is discrete) or a regression function (if the output is continuous). The inferred function should predict the correct output value for any valid input object. This requires the learning algorithm to generalize from the training data to unseen situations in a “reasonable” way.

In our case, we define the output value to be learned as the likelihood of a user to buy in a certain category in the month *m*. The vector of input features is then the set of transactions (or derived data) in the previous months to *m* and demographic attributes. For example, if we consider one year of transactional data, we define the problem as to find the likelihood of a user to buy in the 12th month considering behavioral data from the first month to the 11th month and demographic attributes.

The problem is to compute the **Likelihood** of buying in the category of a Campaign. This function estimates the predicted probability of the user $u_{j}$ to buy in the category $cat(o_{i})$ of the Campaign $o_{i}$ proposed by the merchant $m(o_{i})$ in the following *k* months in which the Campaign will be active (noted as $k=\tau (o_{i})$ henceforth):$$LK\left(u_{j},m\left(o_{i}\right),\tau \left(o_{i}\right)\right)\to \left[0,1\right]$$

The evaluation of the proposed algorithms has been done with one year of prior purchase history (k=1) and for different categories: *shoes*, *tourism*, *sports*, *cosmetics*, and *restaurants*. The historic data that is considered for each instance of the training set contains the transactional data for the last 12 months prior to the month to be predicted.

### 5.1. Classification vs. Regression

Since we aim at computing the likelihood of a user to purchase in a category, a priori, a regression function is better suited. In this case, the output of the inferred function is the probability of a user to buy in a certain category. However, it is also worth exploring the classification approach to determine whether a user would buy in a certain category. A subfamily of classification algorithms, called *probabilistic classification*, outputs the probability of an instance to belong to each of the possible classes (in our case whether the user will buy or not).

We have analyzed the two approaches (classification and regression) to determine which one is better suited to our specific problem and data.

### 5.2. Baseline Algorithm

The Baseline Algorithm will be used to produce evaluation results to be improved by other more sophisticated algorithms. Intuitively, this algorithm is based on the idea that the purchase behavior for the next months will be the same as the past months. It just takes the percentage of months where the user bought in the category as the likelihood of buying in $\tau (o_{i})$.

### 5.3. Linear Regression Algorithm

Linear Regression models the relationship between a dependent variable and one or more explanatory variables. In our case, we have analyzed simple linear regression.) with three different explanatory variables:*The number of months* the user bought in the category in the last 11 months. This variant models the relationship between the number of months the user bought in the category with the odds of buying in that category in the following months.*The number of transactions* the user had in the category in the last 11 months. This variant models the relationship between the number of transactions the user had in the category with the odds of buying in that category in the following months.*The total spent* by the user in the category in the last 11 months. This variant models the relationship between the total money spent in the category in the last months with the odds of buying in that category in the following months.

We also analyzed multiple linear regression with a combination of the three explanatory variables together. Linear regression evaluation results are discussed in Section 5.6.2.

The implementation of this algorithm for our experimentation was done using the statistics package of the *Apache Commoms* Java library.

### 5.4. Logistic Regression Algorithm

In our case, the categorical variable to predict is a Boolean variable indicating whether the user will buy or not in the category in the following months. The output is given as a probability to belong to one class or the other, which is directly interpreted as the odds of the user to buy in a category. We have analyzed the logistic regression approach with the same explanatory variables as in the case of linear regression. Logistic regression evaluation results are discussed in Section 5.6.2.

Other ML models that we tried are Decision Trees, and Random Forests. In the following, we evaluate the performance of the different models for our problem.

### 5.5. Evaluation

In order to find out which of the algorithms we have analyzed perform better, we need a fair and objective evaluation mechanism. To evaluate an algorithm in Machine Learning, a *training set* is defined and used to learn the inferred function. An *evaluation set* is then used to evaluate the performance of the algorithm. The training set and the evaluation set need to be disjoint. We randomly select 90% of the users for the evaluation set, and the rest is used as an evaluation set. We have tested all the proposed algorithms with the same training set and evaluation set so that results can be compared. The evaluation consists of comparing the predicted values in the evaluation set with the actual values. In the following subsection, we enumerate two different metrics to be applied in the evaluation set.

### 5.6. Dataset

We did evaluate the proposed methods with a dataset containing payment transactions from 150,000 customers at the bank for 12 months. Each transaction has the customer Id, the merchant Id, the date, the amount, and the MCC (Merchant Commercial Code), which is mapped to a category in our algorithm. The transactional payment data does not include each product bought within a merchant, but only the total amount spent for a given transaction. Thus, we are predicting how much a customer could spend in a certain category. The transactional payment dataset includes all transactions from customers in a 12 months period, so that the algorithms can consider patterns that are related to seasonality factors.

#### 5.6.1. Evaluation Metrics

We evaluate the proposed algorithms with two different metrics used in supervised learning algorithms:Root Mean Square Error (RMSE). Let be $\bar{x_{i}}$ the inferred probability for a user $u_{i}$ in the evaluation set, and $x_{i}=1$ if $u_{i}$ bought in the *m* month, and 0 otherwise. Then, RMSE for an evaluation set of *n* users $U=(u_{1},...,u_{n})$ is defined as:
$$RMSE_{U}=\sqrt{\frac{\sum_{i=1}^{n}{\left(x_{i}-\bar{x_{i}}\right)}^{2}}{n}}$$Precision and Recall. For classification methods, including information retrieval tasks, Precision and Recall are two evaluation measures commonly used to estimate the quality of a classifier. We need first to define the following:-TP as the number of **true positives**, which are users that are classified as buyers for month *m* in a correct way (they actually were buyers).-FP as the number of **false positives**, which are users that are classified as buyers for month *m* in a wrong way (they actually were not buyers).-FN as the number of **false negatives**, which are users that are classified as non-buyers for month *m* in a wrong way (they actually were buyers).-TN as the number of **true negatives**, which are users that are classified as non-buyers for month *m* in a correct way (they actually were not buyers).Then, Precision (also called Positive Predictive Value) and Recall (also called Sensitivity) can be defined as follows:
$${\mathrm{Precision}}_{U}=\frac{TP}{TP+FP}\;{\mathrm{Recall}}_{U}=\frac{TP}{TP+FN}$$*Precision* measures the fraction of users that have been correctly classified as buyers with respect to all users that have been classified as buyers. *Recall* measures the fraction of users that have been correctly classified as buyers with respect to all users that are indeed buyers.

#### 5.6.2. Results

As stated before, we have evaluated the proposed algorithms, each with different explanatory variables. In Table 1, we show the best performing result from each. According to these results, we elected to proceed with the Random Forest algorithm, as it performed significantly better than the others.

### 5.7. Predicting Purchase Behavior at the Merchant Level

We have studied how to predict purchase behavior at the category level (for example *shoes*). Another interesting factor would be to compute the likelihood of a retail customer to purchase in a specific merchant. We have not explored this venue deeply because the impact on the generic algorithm described in Section 4 would not be very relevant because the algorithms already includes factors such as location (proximity), or merchant loyalty.

However, we also believe this is an interesting metric that can be computed based on functions that have been described in this document. We define a function $LK(u_{j},m\left(o_{i}\right),\tau \left(o_{i}\right))$ that estimates the predicted probability of the user $u_{j}$ to buy in the merchant of the Campaign $o_{i}$ proposed by the merchant $m(o_{i})$ in the following *k* months in which the Campaign will be active as follows:$$LM\left(u_{j},m\left(o_{i}\right),\tau \left(o_{i}\right)\right)=LK\left(u_{j},m\left(o_{i}\right),\tau \left(o_{i}\right)\right)\cdot LY\left(u_{j},m\left(o_{i}\right)\right)$$
where LK and LY were defined in Section 4.2.1.

### 5.8. Algorithm Evaluation for Selected Industries

We have compared the baseline algorithm with the random forest algorithm to predict the likelihood of buying in a given industry for several selected industries: *gastronomy*, *shoes*, *sports*, *cosmetics*, and *tourism*. The results are shown in Table 2 proving that the random forest algorithm works significantly better than the baseline.

### 5.9. Algorithm (Linear and Logistic Regression) Evaluation for Different Variants

Figure 2 and Table 3 show the evaluation of the algorithms based on the number of months the user bought shoes in the previous 11 months, Figure 3 shows the evaluation of the algorithms based on the number of transactions of shoes the user had in the previous 11 months, and Figure 4 shows the evaluation of the algorithms based on the total of money spent in shoes in the previous 11 months. In Figure 2, Figure 3 and Figure 4 the red line charts are depicting the observed data, thus the aim of the algorithms are to approximate the best as possible the function shown as a red line. In all the variants, simple linear regression works better than the other algorithms. Note that in Figure 4, simple regression was applied on the log of the total of money spent in shoes in the previous 11 months. This variable works better in its log form to model user behavior so that small amount differences are more relevant for small numbers than large numbers. Figure 5 and Figure 6 show the errors of the multiple linear regression when predicting the probability of a customer to buy in the shoes category for the following month.

## 6. Conclusions

This paper aims at showing how to build an innovative marketplace within a banking system. It is a full description of the development of a real example on how to apply machine learning to leverage transactional data.

In this paper, we have proposed and analyzed a set of metrics and an algorithm to be able to build a system that matches commercial offers proposed by merchants to retail customers. This matching algorithm includes factors such as the level of accomplishment of each of the active offers, commercial interests defined in the offers, and behavioral and demographic data from retail customers (see Section 4). The proposed algorithm is based on a linear combination of different factors that can be weighted as appropriate. One of those factors is computed through a machine learning algorithm (random forest) that predicts the likelihood of a retail customer to buy in a certain category in the following months (see Section 5).

The system is implemented in a Hadoop platform to be able to process large amounts of data and use machine learning algorithms with Map-Reduce processes (see Appendix A). The system has been deployed and fully tested at a Swiss bank infrastructure, proving that all the processes with real data workflows are working in a real production type of environment. The performance of the system has been proven to be adequate for all the offline processes.

## 7. Future Work and Limitations

This system has been designed for retail small merchants selling at a physical store. However, consumers are increasingly buying online. Online merchants should have another type of analysis since they are selling items in different categories (unlike physical stores). Moreover, for this type of merchants, the concept of proximity should be reviewed. One possibility would be to consider time to deliver in a similar way as we have defined the Proximity factor for physical stores.

Another aspect to be further analyzed could be the seasonality of the purchase behavior. In this regard, it would be interesting to model spending behavior in a natural year and consider how Commercial Interests are affected by different periods of the year.

Further work may include other machine learning methods to compute the likelihood of a customer to buy in a specific category. We have tested linear regression, decision tree and random forest because we were looking for explainability. However, we believe it could be interesting to test other machine learning methods and see how they perform. It is well known that the performance of the learning models come at the price of losing explainability. However, it may be interesting to see how other methods compare to the ones tested in this paper.

## 8. Patents

This work has been patented [8,9] by Strands, Inc., Barcelona, Spain (https://www.strands.com) on 17 December 2014. The patent was published by the US Patent office under the reference US20140365314A1, and in the European Patent office under the reference EP2813994A1. Inventors of the patent are Marc Torrens, Ivan Tarradellas, and Jim Shur.

## Figures and Tables

**Figure 1 entropy-24-00347-f001:**
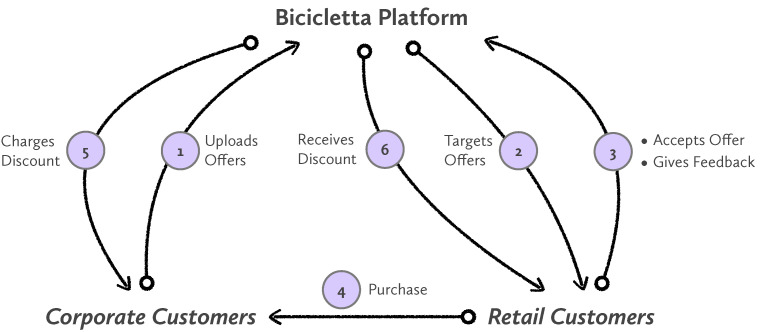
Overview of the system depicting how corporate and retail customers interact with it. The system is comprehensive being in control of all the steps from tailoring offers to the redemption transactions.

**Figure 2 entropy-24-00347-f002:**
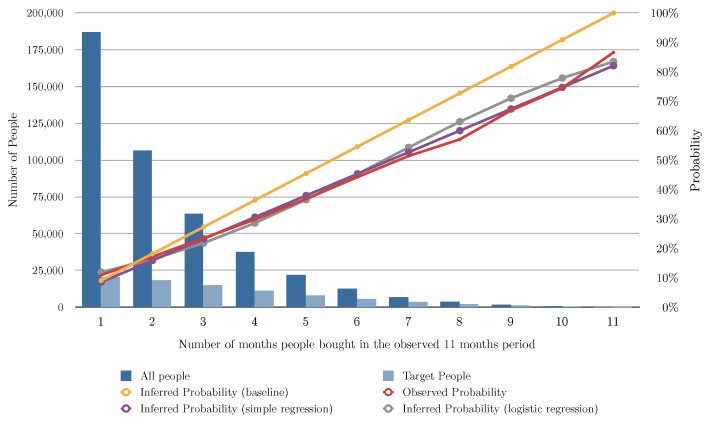
This figure depicts the results of the different analyzed algorithms based on the number of months the user bought shoes in the past 11 months. The horizontal axis represents the number of months the customer bought in the 11 months period. The vertical axis represents the number of people (on the left) for the blue bars, and the probability of buying in the next month (on the right) for the line charts. The dark blue bar indicates how many people bought for each number of months, and the light blue bar indicates how many of those actually bought shoes in the following month. The red line chart indicates the actual probability from the transactional data. This is the function we are aiming at predicting. The yellow line chart indicates the inferred probability using our baseline algorithm (see Section 5.2). The violet line chart indicates the inferred probability using simple regression (see Section 5.3). The gray line chart indicates the inferred probability using logistic regression (see Section 5.4).

**Figure 3 entropy-24-00347-f003:**
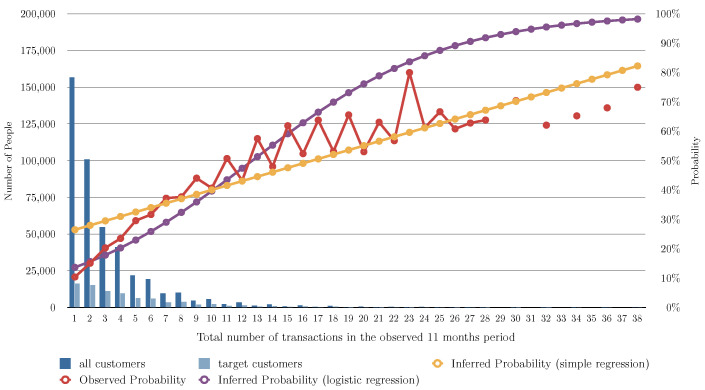
This figure depicts the results of the different analyzed algorithms based on the number of transactions of shoes in the past 11 months. The horizontal axis represents the number of transactions of shoes in the 11 months period. The vertical axis represents the number of people (on the left) for the blue bars, and the probability of buying in the next month (on the right) for the line charts. The dark blue bar indicates how many people have a number of transactions, and the light blue bar indicates how many of those actually bought shoes in the following month. The red line chart indicates the actual probability from the transactional data. This is the function we are aiming at predicting. The yellow line chart indicates the inferred probability using our baseline algorithm (see Section 5.2). The violet line chart indicates the inferred probability using simple regression (see Section 5.3). The gray line chart indicates the inferred probability using logistic regression (see Section 5.4).

**Figure 4 entropy-24-00347-f004:**
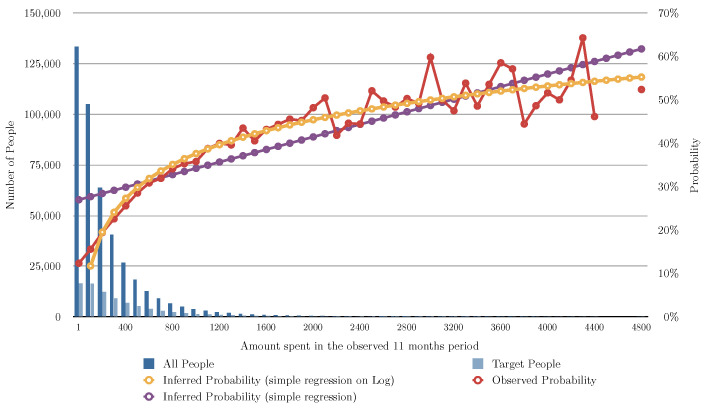
This figure depicts the results of the different analyzed algorithms based on the amount spent by the user in shoes in the past 11 months. The horizontal axis represents the money spent in shoes in the 11 months period. The vertical axis represents the number of people (on the left) for the blue bars, and the probability of buying in the next month (on the right) for the line charts. The dark blue bar indicates how many people spent that amount of money in shoes, and the light blue bar indicates how many of those actually bought shoes in the following month. The red line chart indicates the actual probability from the transactional data. Note that for some number of transactions we do not have any person (missing values). This is the function we are aiming at predicting. The yellow line chart indicates the inferred probability using our baseline algorithm (see Section 5.2). The violet line chart indicates the inferred probability using simple regression (see Section 5.3). The gray line chart indicates the inferred probability using logistic regression (see Section 5.4).

**Figure 5 entropy-24-00347-f005:**
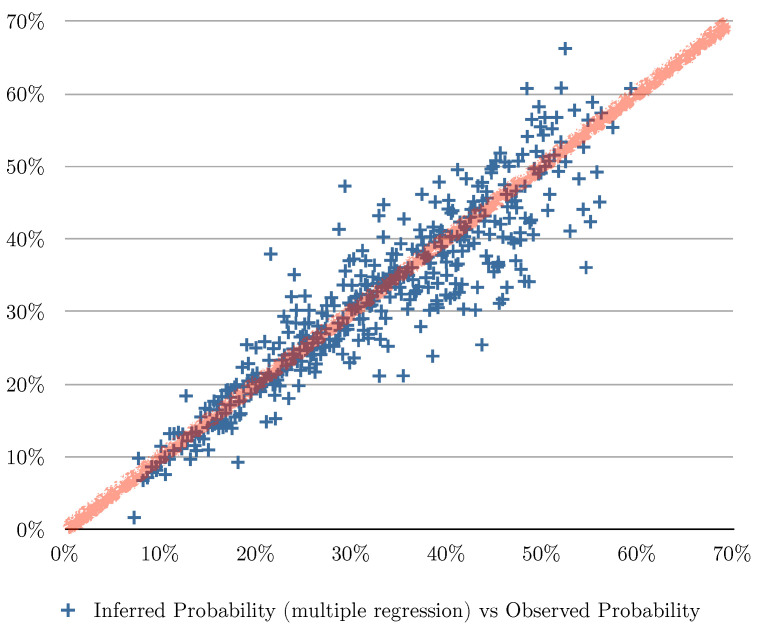
This figure depicts the errors of the predicted values using multiple regression algorithm based on three explanatory variables: (1) the number of the months the customer bought shoes, (2) the total of transactions in shoes, and (3) the money spent in shoes for the last 11 months. For all the instances of the evaluation set, we plot a point in the chart where the vertical axis is the inferred probability and the horizontal axis is the observed probability. The red line indicates what the perfect matching would look like (predicted values equal to observed values).

**Figure 6 entropy-24-00347-f006:**
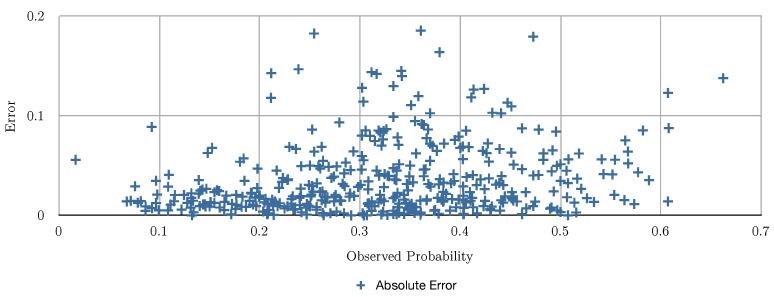
This figure depicts the errors obtained for the multiple regression algorithm.

**Table 1 entropy-24-00347-t001:** Evaluation of different algorithms.

	Baseline	Linear Regression	Logistic Regression	Decision Tree	Random Forest
RMSE	0.3810	0.3783	0.3787	0.3793	**0.3337**
Precision (0.5)	53%	60%	59%	62%	
Precision (0.6)	59%	67%	66%	66%	
Precision (0.7)	65%	71%	73%	74%	

**Table 2 entropy-24-00347-t002:** RMSE results for several selected industries comparing random forest algorithm with the baseline algorithm. We can observer that the random forest algorithm performs better than the baseline in all the selected industries.

	Shoes	Gastronomy	Sports	Cosmetics	Tourism
Baseline	0.3616	0.4000	0.3611	0.3166	0.4316
Random Forest	**0.3524**	**0.3888**	**0.3521**	**0.3054**	**0.3993**

**Table 3 entropy-24-00347-t003:** Evaluation for different variants of linear regression compared to the baseline algorithm.

	Baseline	MONTHS	TXNS-LOG	SPENT	TXNS-SPENT	MON-SPENT	MON-TXNS	MON-TXNS-SPENT
RMSE	0.3810	0.3793	0.3823	0.4016	0.3824	0.3791	0.3783	0.3784
Precision 50%	53.41%	58.93%	65.44%	58.54%	61.83%	58.92%	59.77%	60.71%
Precision 60%	58.93%	64.56%	61.07%	59.26%	65.75%	71.33%	67.20%	67.68%
Precision 70%	64.56%	78.16%	80.56%	50.00%	67.40%	76.34%	71.19%	75.77%

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
