# Peer review of "A Banking Platform to Leverage Data Driven Marketing with Machine Learning"

_entropy, 2022, doi:10.3390/e24030347_

Round 1

Reviewer 1 Report

MDPI Peer Review

Overall Takeaways:

  • This is a well-thought-out, excellent paper with unique and novel application of machine learning in the field of marketing, in a way that has significant real world applications to financial institutions, such as model is likely to be of significant interest to the general reader and public, particularly in the marketing and banking industry
  • The authors have done significant work in development of such a model with relevant testing and implementation
  • At times the paper was difficult to read made in part by the fact that, at times capitalization and punctuation seems inconsistent throughout the paper; would recommend reviewing or having a grammar checker review; for example, the authors appear to use Capital Letters to define key terms but then do not capitalize certain words or terms, implying that they are not defined “Increase Frequency opportunity”, recommend “Increase Frequency Opportunity” or similar
  • Furthermore there are commas / punctuation marks missing throughout the text that make reading difficult to the English reader

Introduction

Overall excellent well thought out explanation of the model and thesis of the paper; but with little reference to other work done in the past in this area. Would recommend further investigation of other work done in this area in addition to further references of similar models developed in the past. Please consider recent work published in the Journal of Big Data, Food Quality and Preference, Applied Artificial Intelligence and Big Data Research which have applied similar techniques in other fields.

Methodology

  • The Proximity component of user to Merchant is an important criteria, but would consider clarification that this is a geographic proximity component compared to a relative proximity component, with a decay component. For example, consider a consumer in London or New York city with a high population density compared to a consumer in rural Idaho or Wyoming with a less dense population. A relative proximity to a merchant in Wyoming may be the closest nearest merchant but similarly penalized to a distant merchant in New Jersey compared to a New York City consumer, whereas a consumer on the UES may be much less likely to purchase from a Brooklyn Williamsburg merchant, depending on travel and physical vs online purchase habits. 

Further explanation of the root data used, for example, how was the determination made for the data used / collected by specific industry, namely for gastronomy, shoes, sports, cosmetics, and tourism. Does this match with the data used for the Purchase Segmentation section? Was there any work done for segmentation, K-means clustering, or hyperparameter tuning of the model? Would touch in in this section. 

  • Conclusions

Good overall discussion of implementation and conclusion in the body of the paper suggesting the relevance and applicability of the model. That said, there is limited discussion of limitations and further work. The fact that a model has been implemented shows that the model has practical and applicable real-life applications but does not in itself show validation of the model. Is there a process to re-evaluate the efficacy of the model or switch between the best regression model? Would clarify. If implementation is discussed, would mention governance and further testing and development.

Lines 539-541: “The system has been deployed and fully tested at a Swiss bank servers proving that all the processes with real data workflows are working in a real production type of environment.” - Is the model deployed and established at the Swiss bank or rather within the server? Would clarify here.

 Limitations/ Further Work

  • Further iterations of the model could compare or weight for population density by zipcode, or determine a proximity based on “time to delivery” of the merchant compared to physical proximity; in addition to consideration of weighting by timeframe, with consideration of seasonality in consumer habits given a 12 month look-back period is applied
  • Other limitations of the data and machine learning framework are not discussed by the authors and should be touched upon, including the fact that there could be bucketing of consumers into different groupings, high volume and low volume “buyers”, similar to higher profile consumers or intermediate consumers - say for example, a local small business or a consumer who is selling sneakers from his apartment to the public, and thus may have a much different spending behaviour

Appendix:

  • Punctuation missing through-out, for example, “Once we have built the model we can make the predictions for all retail customers in the system. Again Mahout is used to apply the learned model.” should have been written “Once we have built the model, we can make the predictions for all retail customers in the system. Again, Mahout is used to apply the learned model.”
  • This sentence also begets the question of whether a version of the model has been completed or a larger model is being built and deployed. Would clarify here.

Author Response

Dear reviewer,

Thank you very much for your detailed review, we appreciate it. Please find our response to your comments:

1) We are going to review carefully the English, grammar, and make the consistent with respect to naming, capitalisation, etc.

2) We are going to add more context with the appropriate bibliography, so the that the paper gets better.

3) We will explain better the proximity component, and the different consumer sectors that were used.

4) We will also add more explanation on the Limitations/Future Work of the system.

Best Regards,

Marc and Amir 

Reviewer 2 Report

Torrens, M.; Tabakovic, A. A banking platform to leverage data driven marketing with Machine Learning. Entropy 2021

The paper proposes an algorithm able to build a system that matches commercial offers proposed by corporate customers to retail customers and consumers within an online banking environment. The paper is interesting but too long.

1). Why the paper is interesting for the audience of the journal, Entropy?

2). My major concern in this exercise is the potential conflict of interest and if there are any regulatory restrictions. I would recommend the authors to discuss this important issue.

3). The paper is too technical, and its objective is not clear.

Author Response

Dear reviewer,

Thank you very much for your comments. Let me answer your comments:

1) We believe the paper is interesting for the special issue on "Machine Learning Ecosystems: Opportunities and Threats" since it presents a full system in production that uses ML to innovate by offering a new business model to banks. There is a large gap between algorithm development "in the lab" and actual systems that work in production. The paper aims at arguing how to build an innovative system in the real world using ML. We think it is important for researchers to think about how to transfer research into "real world" applications.

2) Yes, you are right. From the regulatory point of view, the project had to be carefully analysed with the Swiss authorities. We will add a discussion on that topic. Basically, the conditions to meet regulations were on the direction of transparency for the consumers and total control over the system by customers. So, customers were in full control of the system with a very active role. We will describe how we did it.

3) We agree the paper is long. We are thinking about removing the software architecture part, or add it as an appendix. We would be happy to discuss this with the rest of reviewers and have the best outcome.

We are also improving the introduction, abstract and conclusions so that the overall aim of the paper is better described.

Best Regards,

Marc and Amir 

Reviewer 3 Report

The authors present a new method of data analysis to tailor selling campaigns based on a transactional history of bank customers. The problem is important from the practical point of view. However, considering the Entropy audience there is a question about the scientific impact of the paper. A proper introduction with a necessary literature review should be provided.

The next point is the machine learning method choice. The authors have tested a few of them, but without necessary information why those. This branch is developing very quickly and there are a lot of methods so at least some arguing upon the method choice is welcomed.

The analysis does not cover the Internet companies, in which case the distance is a less important factor, but the transportation costs and delivery conenience.

Mentioning the distance there are terrible mistakes in units (l. 353 - 356). The distance unit is "km" not "Km".

The last point is the question of why this paper should be published in the Entropy journal. Let remind that the journal scope is:

  • develop the theory behind entropy or information theory
  • provide new insights into entropy or information-theoretic concepts
  • demonstrate a novel use of entropy or information-theoretic concepts in an application
  • obtain new results using concepts of entropy or information theory

Author Response

Dear Reviewer,

Thank you very much for taking the time to review our paper. We are definitely improving the literature review within the Introduction section, as you suggest. We also agree that we should justify why we have selected those machine learning methods, so we are going to add an explanation for that (around explainability). Also, we are going to mention the distance metric should be adapted to Internet companies.

The reason why we are submitting this paper to the special issue on "Machine Learning Ecosystems: Opportunities and Threads" is that it presents an overall view on a real-case application that uses machine learning. It aims at exploring how to bring a machine learning based system into production. We hope to have completely addressed your concerns that will be addressed in the reviewed paper.

Thank you again for your consideration.

Best Regards,

Marc Torrens

Round 2

Reviewer 1 Report

The Authors have made the relevant edits and changes to the manuscript based on review. The updated manuscript reads well and incorporates suggested changes.

Author Response

Dear reviewer, thank you very much for your comments. We are very happy to see the paper improved based on your comments.

Best regards,

Marc Torrens

Reviewer 2 Report

Dear Authors,

Thank you for the revised version of your manuscript. 

Author Response

Dear reviewer, thank you very much for your work reviewing the paper. We are very happy to see our paper improved based on your comments.

Best Regards,

Marc Torrens